# Post-FSW Cold-Rolling Simulation of ECAP Shear Deformation and Its Microstructure Role Combined to Annealing in a FSWed AA5754 Plate Joint

**DOI:** 10.3390/ma12091526

**Published:** 2019-05-09

**Authors:** Marcello Cabibbo, Chiara Paoletti, Mohamed Ghat, Archimede Forcellese, Michela Simoncini

**Affiliations:** 1Dipartimento di Ingegneria Industriale e Scienze Matematiche (DIISM), Università Politecnica delle Marche, Via Brecce Bianche 12, 60131 Ancona, Italy; c.paoletti@univpm.it (C.P.); m.ghat@univpm.it (M.G.); a.forcellese@univpm.it (A.F.); michela.simoncini@uniecampus.it (M.S.); 2Università degli Studi eCampus, Via Isimabrdi 10, 22060 Novedrate, Italy

**Keywords:** FSW, CR, ECAP, TEM, hardness, AA5000 series

## Abstract

Friction stir welds are considered reliable joints for their lack of voids, cracks and distortions. When compared to the base material, friction stir welding (FSW) joints typically exhibit finer grain structured (especially at the nugget zone, NZ). Similarly, refined grain structure can also be obtained by severe plastic deformation (SPD) techniques, such as equal channel angular pressing (ECAP). In fact, the fine grain structures produced within the NZ of FSW or friction stir processed (FSP) materials are usually coarser than the ones achieved by ECAP. The former is characterized by lower dislocation density, higher high-angle boundary fraction and different mechanical strength, compared to what can be obtained by ECAP. In this study, a dedicated cold-rolling (CR) set-up, specifically designed to simulate an ECAP-equivalent shear deformation, was used to further refine the grain structure of FSW AA5754 sheets. The effect of ECAP-equivalent deformation induced by CR in a 2 mm-thick AA5754-H111 FSW joint was investigated. FSW was carried out at two different rotational (*ω*) and translational (*v*) welding speeds, 600 rpm, 200 mm/min and 1800 rpm, 75 mm/min, respectively. FSW sheets were then CR to obtain an equivalent shear strain of *ε* ~ 1.08, that is equivalent to 1-ECAP pass carried out with an internal die channels intersecting at an angle *φ* = 90° with a curvature extending over an angle *Ψ* = 20°. By CR, the sheet thickness reduced only by ~20%. The role of annealing on the FSW and CR plastically deformed AA5754 was also investigated. This was applied either prior or after FSW, and it resulted that whenever it follows the FSW, the mean volume fraction of dispersoids and Mg-rich particles is higher than the case of annealing preceding the FSW process. On the contrary, it was found that the annealing treatment had a minimal role on the dispersoids and particles mean size. The here reported post-FSW ECAP-simulated deformation, obtained by a customized CR process, showed sheet integrity and a significant concurrent grain size refinement.

## 1. Introduction

Friction stir welding (FSW) is a solid-state technology developed to obtain joints with mechanical properties potentially higher than the ones usually obtained by conventional fusion welding techniques. FSW is acknowledged as an effective joining technology for lightweight materials, such as aluminum alloys, that are usually difficult to weld, or even unweldable by fusion technologies. Nowadays, FSW is applied to several aluminum alloy systems for a variety of industrial application, from automotive, to aerospace, to leisure and technology components [1,2,3]. FSW sheet blanks are joined by the action of a rotating pin tool getting in contact with the top surface of the sheets, and then by moving the pin along the welding line with a set tilt angle. The combined effect of the plunging, the rotation, and the translation of the welding tool generates friction heating between the tool and the sheets. The process thus induces the material plastic deformation by stirring and promotes a complex mixing across and along the joint path. This way, a plasticized region is created around the tool and beneath the shoulder of the pin. This way, the welding material is extruded from the retreating (RS) to the advancing side (AS) [1,4,5].

During FSW (and also friction-stir processing, FSP), the material that flows around the tool undergoes intense plastic deformation and local temperature rise. Both the mechanical plastic deformation and the thermal excursion (well within the alloy melting point) are normally able to produce a fine-gained structure in the center of the welded zone. This central zone is commonly referred to as the nugget zone (NZ). The occurrence of a fine grain structure in the NZ significantly affects the mechanical properties of the welded/processed alloy. Correlation between the NZ and the outer zones of the welded joint (namely RS and AS) are microstructure aspects of significant technological and scientific importance. Several literature studies have successfully explained the formation of fine grains within the NZ as a dynamic recrystallization-based (DRX) phenomenon. The NZ DRX is likely to occur continuously either by rotation of existing subgrains, or by subgrain formation during the FSW, which eventually are thermally induced to evolve into high-angle boundary network, that is, into a fine grain structure [6,7,8].

To maintain a constant sheet thickness after FSW, the FSW blank can be subjected to a cold rolling (CR) process. On the other hand, CR after FSW increases the alloy strain-hardening coefficient, thus improving both material strength and formability [9]. However, the degree of strain hardening and the location of the welding line, with respect to the rolling direction, are known to affect the mechanical properties and formability of the welded sheet. With this respect, the effect of CR on the mechanical properties and formability of FSW sheets of AA5754 aluminum alloy was already studied and discussed by Casalino et al. in [10], and by other researchers [11,12,13]. In a previously published work on FSW + CR of the same H111-AA5754 [10], it was found that both yield and ultimate tensile strength in the FSW + CR condition were higher than those obtained in the FSW joint. Moreover, the mechanical behavior was shown to be strongly strain-hardening dependent, where most of the strain hardening comes from the post-FSW CR. With this respect, the strain-hardening exponent was found to reduce with hemispherical punch height reduction. Accordingly, the FSW + CR showed a ductile response within the formability limit regime [10].

Based on the observed good mechanical response of the FSW + CR AA5754, the plastic response of such a sequence of deformation was furthermore studied to simulate an equal channel angular pressure (ECAP)-like deformation on FSW AA5754. To this purpose, FSW was performed on 1.5 mm-thick sheets, using a pin tool with constant values of rotation and welding speeds. Strips of the FSW sheet were CRed, with the rolling direction parallel to the FSW line path. A single pass through the CR gage was carried out and the equipment was designed and set to induce an equivalent strain of *ε* ~ 1.08. This equivalent strain is the same that was obtained by one of the present authors, using a ECAP die consisting of two equal channels intersecting at an angle *ε* ~ 90° and a curvature angle *Ψ =* 20° ([14,15,16,17] and references therein). After FSW, the sheet thickness across the NZ was of 1.3 mm, while after FSW + CR was of 1.1 mm throughout the strip.

## 2. Experimental Details and Method

FSW was performed in a machine center on an H111-AA5754 aluminum alloy. The AA5754 alloy had a chemical composition (wt.%) of 3.4 Mg, 0.40 Mn, 0.20 Si, 0.20 Fe, 0.10 Cr, 0.15 (Ti + Zn), and Al bal. The sheet blanks were 200 mm long, 60 mm width, and 1.5 mm thick. Initial grain structure of the AA5754 sheet had elongated grain in the rolling direction whose mean size was 180 ± 60 μm (Rolling direction, RD and Transverse Direction, TD), and a mean thickness of 50 ± 15 μm (normal direction, ND). Sheets were butt joint. A X35CrMoV5-tool steel pin (H = 50 HRC) with a shoulder diameter of 15 mm, a truncated conical shape with a base of 3.9 mm, and a height of 1.3 mm was used. During FSW, the tool was initially forced 0.1 mm down into the aluminum sheet. Welding was carried out with a pin nutting angle equal to 2°. The welding line followed a path perpendicular to the rolling direction (RD) of the as-received sheet. Two different welding parameters (rotational speed, *ω*, and welding translational speed, *v*) were used. One welding procedure was carried out with *ω* = 1800 rpm and *v* = 75 mm/min, a second welding was carried out by *ω* = 600 rpm and *v* = 200 mm/min welding settings. These settings were chosen according to the weldability window of the AA5754 alloy that was determined in two previously published works by some of the present authors [10,18].

CR of FSW sheets was carried out on 25 mm-width and 40 mm-long rectangular strips extracted from the welded sheets along the welding path, having the welding NZ right in the middle of the strip face. The strips were rolled along the previous FSW line (CR direction same of the Welding direction), as shown in Figure 1. The FSW strips were CR once, and the CR machine was set to impose an equivalent strain of *ε* ≅ 1.08. The CR set-up is schematically reported in Figure 1. The scheme of Figure 1b, shows the settings used to achieve the desired equivalent strain. In the present case, the center-to-center distance of the two rolls was ΔR ≅ 13 mm, and both rolls had the same rotational speed of 60 mm/min, the upper rolling counterclockwise, while the bottom one clockwise. With this CR configuration, since the minimum height of the upper roll is at the same level of the maximum height of the bottom roll, the welded sheet is forced to bow as soon as it met the low roll. The sheet is thus forced by plastic shear by the action of the two adjacent rolls at the contact point of the sheet with both the two rolls. This way, the sheet is forced to follow the upper roll curvature, that is, to follow a path of ~90° with respect to the initial horizontal rolling direction. 

The main goal of the present study was to check the feasibility to simulate an ECAP straining in a FSW aluminum weldment, by using a cold-rolling technique. The need for using a CR technique instead of a conventional ECAP die, after FSW, comes from a technological viewpoint. In fact, in this work, the authors aimed at probing the industrial scale up for a combined mechanical process of welding and plastic deformation able to effectively refine the NZ grain structure. The single pass of the strips through the CR gage is specifically designed to impose a strain of *ε* ≅ 1.08 and to simulate a single pass into a L-shaped two-channel die. With this respect, the imposed equivalent strain of *ε* ≅ 1.08 by CR, simulates an ECAP die consisting of two equal channels intersecting at an angle *φ* = 90° with a curvature extending over an angle *Ψ* = 20° [14,15,16,17,19]. On the other hand, the CR set-up here used and above described (see also Figure 1b) had an overall deviation angle of ~90° and a curvature of ~15–20°, which is imposed by the upper roll. To determine the feasibility of the *ε* ~ 1.08 CR microstructure refining process with an equivalent strain induced by ECAP, the above described ECAP die was used to obtain an AA5754 billet plastically deformed at 1-pass (*ε* = 1.08). Thus, a billet of AA5754 was deformed by shearing deformation mode following 1-ECAP pass at room temperature. ECAP was carried out with plunger pressing speed of 40 mm/min and a pressure of 40–45 kN.

After FSW the sheet thickness on the welding NZ was of 1.40 ± 0.05 mm, while after FSW + CR the thickness reduced to 1.2 ± 0.1, that is a reduction of ~20% was imposed by the combination of FSW and CR.

A sequence of fully annealing was also added to the FSW and CR processes. The alloy was fully annealed at 415 °C/3 h and then cooled inside the shut-down furnace. The annealing was done in two different sequences: one, prior FSW + CR (A-FSW-CR), a second, between FSW and CR (FSW-A-CR). The annealing heat treatment was added to the welding-plastic deformation experimental sequence, with the scope of determining the role of FSW and FSW + CR on the MnAl_6_ dispersoids precipitation and on the occurrence of Mg_2_Al_3_ particle precipitation in the AA5754 Al-Mg-Mn alloy. Microhardness was also measured through the different meaningful welded zones (nugget, thermo-mechanical heat affected zone, heat affected zone) in all the experimental conditions here tested, by using a Remet^®^ HX-1000^TM^ tester (Remet, Casalecchio di Reno, Bologna, Italy). Hardness measurements were spaced 250 μm apart and taken along the mid-line section of the welded strips.

TEM discs were prepared by polishing, dimpling and ion-milling techniques. Inspections were carried out using a Philips^TM^ CM-20^®^ (Philips, Amsterdam, Netherlands), equipped with double tilt specimen holder. Two different discs per experimental condition were inspected. Evaluation of grain and cell sizes was made after recognition of the boundary character of some selected disc zone, by using the Kikuchi pattern method. By Kikuchi patterns, the minimum rotation across adjacent boundaries to bring them to coincide can be assessed to a minimum typical value of 0.1°. In particular, the boundary misorientation is determined through a misorientation matrix Rcr = Rcp × Rpr, where Rcp is the matrix of the angles between the crystal and the electron beam and Rpr is the matrix of the angles between the reference crystal and the beam direction. This boundary misorientation method is properly described in two previously published papers by Cabibbo [16,17], and the reader is addressed to [17] for further details.

## 3. Results and Discussion

Table 1 and the table addendum report the experimental thermo-mechanical sequence to which the AA5754 was subjected.

Figure 2 shows an optical microscopy (OM) montage of the alloy after FSW1 (*ω* = 600 rpm, *v* = 200 mm/min) and after CR to an equivalent strain *ε ~* 1.08. No evidence of a crack was observed throughout the welded sheet section (Figure 2a), from the left-hand side, where the retreating side (RS) is located, to the right-hand side, where is located the advancing side (AS). On the other hand, no cracks were observed throughout the top FSW joint surface (Figure 2b), which was the one that experienced the larger plastic stress during CR, as it was the outer surface during CR (see also Figure 1). These representative macros show that the CR process did not cause any microstructure cracking through the RS, the NZ, and the AS. 

Figure 3 reports polarized optical micrograph (POM) montages of the FSW sections of the four different sequences and set-ups of A-FSW-CR processes. Here, from left to right, RS heat affected zone (R-HAZ), thermo-mechanical affected zone (R-TMAZ), NZ, and the corresponding AS A-TMAZ, and A-HAZ can be observed.

Both FSW process parameters (FSW_1_ and FSW_2_), seems showing that the post-FSW annealing significantly increased the DRX fine grain size (Figure 3a,c). On the contrary, when the annealing preceded the FSW, no significant grain grown occurred. Yet, the TEM inspections, reported in the followings, showed that the grain essentially remained fine, that is of few microns in size, also when annealing followed the FSW. As a matter of fact, the here observed coarse areas within the NZ are actually a broad portion of grain structure characterized by boundaries with large misorientation angles. The polarized optical micrograph (POM) montage of Figure 3 essentially shows in all the four experimental conditions here studied no significant cracks formation after CR.

TEM inspections carried out on FSW_2_, FSW_2_ + CR (*ε* = 1.08), and ECAP-1 (*ε* = 1.08, X-plane [17]) revealed grain structure and tangled dislocation differences (as shown by the representative TEM images of Figure 4). In ECAP-1, grains are significantly deformed in the ECAP extrusion direction, high-dislocation density bands were formed oriented to 40°–45° with respect to the pressing direction. The shear plastic deformation induced into the alloy during the ECAP first pass is also responsible for a large tangled dislocation density in the grains and cells interior. These will rearrange and reorganize to form low-angle boundaries, and thus new cells under following passages into the ECAP die [15,16,17,20,21]. On the other hand, the FSW NZ was characterized by an almost equiaxed grain structure containing quite low-density of tangled dislocations within the grains. In this latter case, the grains were formed by a thermo-mechanical stirring process acting in the NZ. This ultimately induced a recrystallization process on the existing alloy microstructure [21,22,23,24,25,26]. This aspect is one of the key deformation mechanisms that differentiate a grain-refinement process induced by friction stir process (FSP) from a grain-refinement generated by ECAP. To some extent, the grain structure of the FSW NZ after CR performed in one single passage to induce an equivalent strain *ε* = 1.08 shows a microstructure similar to both the grain deformation microstructure induced by ECAP-1 and the one obtained by FSW. In fact, the FSW + CR microstructure appeared more equiaxed than that observed in ECAP-1 microstructure. This was also characterized by the presence of bands of high-density of dislocation, that somewhat reminds the typical shear deformation that is induced by ECAP. In the FSW + CR case, the tangled dislocation density, within grains and bands, is surely higher than what found in the FSW NZ. Actually, the tangled dislocation density in the FSW + CR sample was quite similar to what obtained after ECAP-1. 

As for the mean grain sizes, *d_g_*, it was found that after the FSW process the NZ had a grain size *d_g_* ranging 30–60 μm, which was some four times larger than the mean grain size obtained by ECAP-1 (*d_g_* ~10 μm). The ECAP-1 pass mean grain size was slightly larger compared to the FSW + CR mean grain size at the NZ (*d_g_* ranging 6–8 μm). Table 2 reports the mean grain, d_g_, and cell (subgrain), *d_c_*, size for all the experimental conditions. As for the cell structure dimension, *d_c_*, and fraction, *V_v_*, the FSW sample did not show any significant present of low-angle boundary crystallites, as most of the boundaries were generated by a recrystallization phenomenon. On the contrary, a considerable fraction of the existing boundaries in the ECAP-1 condition had a character of low-angle, and thus the cells were abundant within the fine grain structure, whose typical sizes was *d_c_* ~ 1 μm. Cell size, *d_c_*, in the FSW + CR condition was significantly lower that what found after ECAP-1, as in FSW + CR it ranged from 500–800 nm. In this sense, it seems that the combination of FSW and CR performed with a significant amount of plastic deformation to simulate a *ε* = 1.08 equivalent strain had a great potential to promote both grain size refinement and to generate sub-micrometer cell structures. These, in turns, will have the chance to further refine the alloy grain structure upon further plastic deformation. With this respect, to some extent, the ability to effectively refine the grain structure of aluminum alloys, and specifically the AA5000 series alloys, was also explored and documented by some other authors, namely the groups of Kaibyshev and Lewandowska [27,28,29]. It resulted that the here presented results are in line with what was reported by these authors.

The microstructure role of the annealing treatment on the FSW + CR plastically deformed AA5754 was studied both in terms of grain structure, and in terms of secondary phases and Fe-based intermetallic particles. Figure 5 reports representative TEM micrographs of different process sequences: FSW-A-CR and A-FSW-CR for the two experimental FSW set-ups (FSW_1_ and FSW_2_) here analyzed.

Whenever the annealing treatment follows the FSW, it appeared that the mean grain size was smaller than the case when the annealing precedes the FSW. This was observed irrespectively of the FSW parameters (that is for FSW_1_ and FSW_2_). Moreover, the grain morphology was generally more elongated in the FSW-A compared to the A-FSW sequence of the A + FSW + CR processes. On the other hand, the post-FSW annealing of the two FSW-A-CR conditions revealed a larger density of tangled dislocations than the two A-FSW-CR conditions.

Indeed, the A-FSW_1_-CR microstructure showed lower mean grain size than the A-FSW_2_-CR microstructure, as well as the FSW_1_-A-CR condition with respect to the FSW_2_-A-CR one. In fact, the mean grain sizes *d_g_* were 6 ± 1 μm, 1.1 ± 0.4 μm, 2.8 ± 0.6 μm, and 3.6 ± 0.6 μm, in the FSW_1_-A-CR, FSW_2_-A-CR, A-FSW_1_-CR, and A-FSW_2_-CR, respectively. On the other hand, the mean cell size differences with respect to the FSW parameters were less evident, as their mean value d_c_ were 800 ± 100 nm, 1100 ± 200 μm, 500 ± 100 μm, and 600 ± 100 μm, in the FSW_1_-A-CR, FSW_2_-A-CR, A-FSW_1_-CR, and A-FSW_2_-CR, respectively. The size difference was actually due to the FSW post- or pre-annealing treatment, as it reduced by some 30–35% from the post-FSW to the pre-FSW annealing sequence. Grain size distribution of FSW, FSW-CR, and ECAP-1 conditions is reported in Figure 6.

On the other hand, the role of plastic deformation and thermal excursions induced by the passage of the pin and shoulder during FSW, the role of the post-annealing, and post-CR on the microstructure modifications occurring in the NZ was also detected by TEM. Figure 7 reports two representative micrographs documenting the occurrence of dislocation walls and cell (subgrain) formation. In fact, different kinds of dislocation structures were observed in the DRX grains of the NZ. In addition, the post-FSW annealing showed grains with different degrees of recovery next to each other and they were characterized by low tangled dislocation density, compared to the FSW-CR condition. The occurrence of dislocation walls within the grains (Figure 7a) is likely to indicate that a full microstructure recovery was not reached after DRX, in the FSW_1,2_-A-CR conditions. On the other hand, the formation of cell structures within the DRX grains in the NZ, after CR, also denotes that the DRX process was not yet fully completed (Figure 7b). With further plastic deformation, these subgrains are eventually able to grow in their misorientation degree and thus eventually promoted to high-angle boundary (i.e., nucleating new grains). The here observed process of dislocation rearrangement and pile-up was also documented by other researchers ([7,30], to cite but two). 

The DRX-driven formation of fine grains in the NZ starts with dislocation arrangement to form boundaries by the plastic deformation and the thermo-mechanical cycles during FSW. This early process is followed by dynamic recovery taking place by the formation of subgrains with low-angle boundaries. In fact, dislocations are continuously introduced also during the following CR and tend to pile-up in the existing subgrains to reduce the local strain. In this context, the effect and role of the post-FSW annealing is chiefly that of relieving the plastic stress and induce tangled dislocation to further form low-angle boundaries. During CR, the subgrains are induced to grow and absorb the previously formed tangled dislocations into subgrain boundaries. Thus, the formation of cell boundaries during CR is greatly promoted. 

In order to correlate the here observed grain and cell microstructure differences, microhardness profiles were measured along the FSW joint sections in all the experimental conditions. Figure 8 shows microhardness plots across the FSW joint sections of the FSW_1_ setting (on the left-hand side), and of the FSW_2_ setting (on the right-hand side).

As reported in Figure 8, hardness profiles across the FSW joint show a certain degree of uniformity among the different characteristic FSW zones. The FSW_1_ mode showed a higher uniformity than the one where the FSW_2_ parameters are used. In particular, FSW_1_ shows a hardness profile across the retreating heat-affected zone (R-HTZ), the R-TMAZ, the NZ, and the advancing counterparts, quite similar to what observed in the FSW_1_-A-CR experimental condition. The corresponding hardness values were of ~86–90 HV_0.3_ in the NZ. Similar hardness values were obtained in the A-TMAZ and R-TMAZ. The hardness values in the two opposite R- and A-HAZ were lower by some 10%. On the other hand, the hardness profile of the FSW_1_ + CR condition had systematically higher values than the two above-mentioned, peaking to ~100 HV_0.3_ in the A-TMAZ. As for the A-FSW_1_-CR deformation sequence, the hardness profile was significantly lower than the previous three experimental conditions. This also showed a higher uniformity than the other profiles, with hardness values ranging 70–76 HV_0.3_, where the peak values were essentially obtained in the NZ. Thence, the marked effect of the pre-FSW annealing was to slightly soften the hardness throughout the joint section.

The hardness profiles obtained with the FSW_2_ parameters were significantly more scattered and less uniform throughout the FSW joint section, as compared to the FSW_1_ parameters. The hardness value relationship among the different experimental conditions remained essentially similar to what observed for the FSW_1_. Anyhow, the hardness values were constantly higher in all the four tested experimental conditions, compared to the FSW_1_ settings. Thus, a general hardness increment by ~30% characterizes the different FSW zones, from the R-HAZ throughout to the A-HAZ.

With this regard, some other authors [31,32,33,34,35] also studied the role of the FSW parameters on the hardness profiles, and, to some extent, on their influence on the hardness profiles across the FSW joint section after a pre- and post-FSW heat treatment. It is then generally agreed that as the pin transverse speed rises and as the pin rotation spinning rises, the hardening effect rises accordingly. The here presented hardness profiles well agree with this FSW setting parameter influence on the resulting hardness across the joint, as it chiefly rises with the pin transverse speed.

To further correlate the role of the pre-/post-FSW annealing, and that of the final CR, a direct comparison of the mean hardness values, taken at the NZ was made.

The average hardness values across the NZ, reported in the Table 3, showed a common trend between the two FSW processes (FSW_1_ and FSW_2_) in relation to the annealing and CR sequences. NZ hardness increased by 10% from FSW to FSW-CR, in both conditions: FSW_1_ and FSW_2_. On the other hand, the mean hardness reduced to values lower than the ones obtained by FSW, whenever the annealing follows the FSW. This reduces furthermore to ~15% compared to the FSW values, whenever the annealing treatment precedes the FSW. This means that, as expected, the hardening effect of post-FSW CR is to slightly raise the hardness in the NZ, while the annealing treatment is responsible for a hardness reduction, which is more pronounced in the pre-FSW condition.

The comparison between the FSW, FSW-CR, and ECAP-1 (with same plastic deformation rate of *ε* = 1.08, than that induced by CR) shows that the hardness achievable by the FSW_2_ and FSW_2_ + CR is only 5% and 15% higher, respectively. This mechanical results and the microstructure evidence of very fine grain obtained in the NZ imply that is actually possible to achieve same, or similar, effects by using FSW + CR than those usually achieved by ECAP-1 (*ε* = 1.08).

The role of the annealing treatment introduced in the FSW + CR sequence was inferred in terms of statistical evaluation of MnAl_6_ dispersoids, Mg_2_Al_3_ particles, and, to some extent, (Fe,Mn)_3_SiAl_12_ intermetallic particles. This was carried out by TEM analytical inspections.

Figure 9 reports representative TEM micrographs of the FSW_1_-A-CR, A-FSW_1_-CR, FSW_2_-A-CR, A-FSW_2_-CR, and ECAP-1 conditions, while Table 4 shows the statistical data of the particle volume fraction, *V_v_*, and mean size, *d_p_*.

One of the major and most interesting aspect of the deformation sequence induced by FSW and following CR is surely the Fe-rich intermetallic cutting process that was promoted by the combination of plastic deformation of, first, stirring, and, after, shearing, type.

The intermetallic particle shearing phenomena was also observed in the sample ECAP-1 (Figure 9e), and was also documented and quantitatively calculated in a different case by one of the present author (M. Cabibbo), in two previous published works on Al-alloys [36,37].

This means that, the combined FSW and CR plastic deformation has a similar effect and influence on the Fe-rich needle and rod-like particles than the ECAP, which actually induces a pure shear-type plastic deformation into the alloy grain structure.

On the other hand, the clear effect of the annealing treatment, prior or after FSW is that, whenever it follows the FSW, the mean volume fraction of both the MnAl_6_ dispersoids and the Mg_2_Al_3_ particles is higher than the case of annealing preceding the FSW process. In particular, the Mg_2_Al_3_ particles volume fraction was the lowest in the FSW_2_ condition, *V_V_(Mg_2_Al_3_)* ~8‰, and increased to be the maximum, among the inspected experimental conditions, when annealing and CR followed the FSW_2_, *V_V_(Mg_2_Al_3_)* ~28‰. Correspondingly, also the MnAl_6_ dispersoids increased in volume fraction from the FSW_2_ to the FSW_2_-A-CR condition, accounting for a 60% rise, i.e., *V_V_(Mg_2_Al_3_)* ~50‰ to ~80‰, and the Mg_2_Al_3_ particle *V_V_(Mg_2_Al_3_)* reached a three-fold increment. These particle increments were essentially due to the combined effect of post-FSW thermal energy, on annealing, and shear-type plastic deformation during CR.

In all the experimental conditions, the (Fe,Mn)_3_SiAl_12_ intermetallic particle mean size, d_p_((Fe,Mn)_3_SiAl_12_) ranged (300 ± 50)· (600 ± 30)· (1000 ± 80) nm^3^, the volume fraction V_V_((Fe,Mn)_3_SiAl_12_) ranged (130 ± 40)‰. 

Another interesting result, coming from the quantitative measurement of the MnAl_6_ particle *V_V_(MnAl_6_)* and mean size *d_p_(MnAl_6_)*, is that both did not seem to be affected by the specific sequence of the annealing, as their appeared to be quite similar in either A-FSW_1,2_-CR and FSW_1,2_-A-CR conditions. This was not the case for the Mg_2_Al_3_ particles, as these were found to reduce in volume fraction by at least 30%, to even 50% by inverting the annealing sequence, from post- to pre-FSW. On the other hand, the *d_p_(MnAl_6_)* was not affected by the different thermo-mechanical deformation that was followed in the present study.

Thus, to some extent, the annealing treatment influenced the dispersoids and particles fraction but had a minimal, if not, irrelevant role on their mean size.

## 4. Conclusions

In the present study, a plastic deformation process of cold rolling (CR) on a FSWed H111-AA5754 sheets was tested with the goal of obtaining a grain refinement as similar as possible to the one obtained by a simple equal channel angular pressing (ECAP) single passage into the die (ECAP-1). To this purpose, a dedicated CR set-up, able to plastically deform the FSWed sheet to an equivalent strain of *ε* ~1, and inducing a ~90° deformation, was designed. The study also focused on the role of the annealing treatment, to which the alloy was subjected either prior or after the FSW, but always before CR. The FSW + CR induced microstructure modification was compared to the microstructure obtained by ECAP-1, which induced an equivalent shear strain of *ε* = 1.08. 

It was thus found that:The grain size potential of coupling FSW and CR was quite similar to what was obtained by the same strain level ECAP-1. This similarity holds not only for the resulted microstructure, but also in terms of the obtained hardness increment.The hardness influence of the annealing treatments was detected in the FSW nugget zone (NZ). It was found that, whenever the annealing treatment preceded the FSW, the annealing treatment significantly reduced the hardness, compared to the FSW + CR hardness. On the other hand, in the FSW-A-CR sequence, the hardness of the post-FSW annealing was similar to what was obtained just after FSW.As for the annealing treatment influence on the MnAl_6_ dispersoids and Mg_2_Al_3_ particles, this was determined in terms of their mean size (*d_p_*) and volume fraction (*V_v_*). It resulted that in the case of the MnAl_6_ dispersoids, *d_p_(MnAl_6_)* was not affected by the sequence of the annealing, i.e., pre-/post-FSW. The *V_v_(MnAl_6_)* increased in either cases of pre- and post-FSW annealing. In the case of the Mg_2_Al_3_ particles, *d_p_(Mg_2_Al_3_)* was not significantly affected by the combined effect of annealing and CR, while the *V_v_(Mg_2_Al_3_)* increased preferentially when annealing followed the FSW, compared to the other way round.The obtained results of MnAl_6_ dispersoids and Mg_2_Al_3_ particles evolution with A-FSW-CR and FSW-A-CR were quite in line with what was obtained by ECAP-1.The here obtained results seem to confirm the possibility to design a scale up ECAP process dedicated to FSW joint aluminum sheets.

## Figures and Tables

**Figure 1 materials-12-01526-f001:**
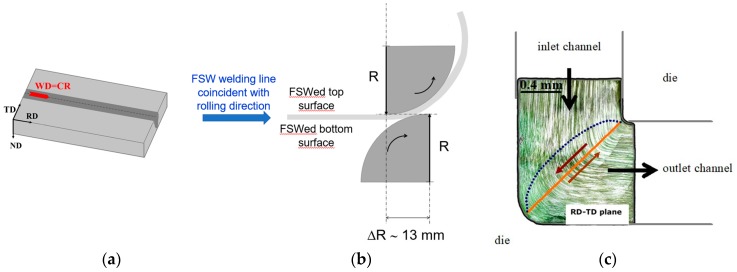
Sheet Rolling Direction, RD, Welding Direction, WD, and related RD, transverse direction, TD, and normal direction, ND, (**a**) in scale scheme of the cold rolling set-up designed to induce an equivalent strain of *ε* = 1.08, and a ~90° rotation of the friction stir welding (FSW) joint strip under a single sample passage through the gage, (**b**) The induced plastic deformation aimed to simulate a typical 90° shear deformation achievable by equal channel angular pressure-1 (ECAP-1) pass, as shown schematically in (**c**).

**Figure 2 materials-12-01526-f002:**
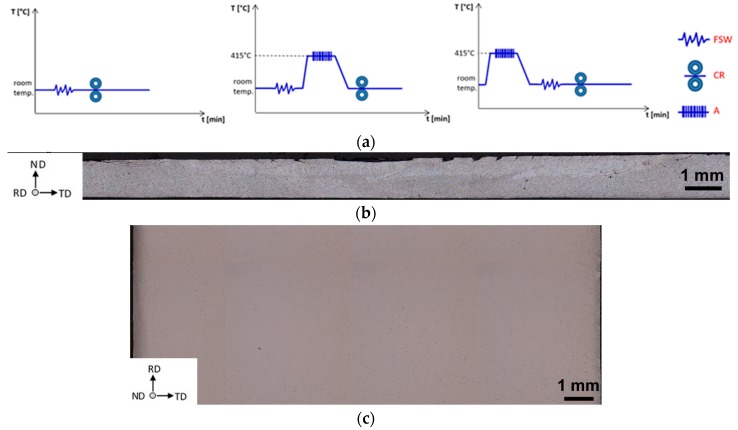
Schematic representation of the thermo-mechanical processes. (**a**). Optical microscopy (OM) of the FSW_1_ (*ω* = 600 rpm, *v* = 200 mm/min) after annealing at 415 °C/3 h (A) and cold rolling (CR) to ε ~ 1.08 (FSW_1_-A-CR), of the AA5754 sheet section, where the retreating side (RS) is on the left-hand side, and the advancing side (AS), on the right-hand side, (**b**). Central region top surface, corresponding to the nugget zone (NZ), i.e., the bottom FSW surface, corresponding to the one CRed at the top roll cylinder, (**c**). Both lateral section and top FSW-CR surface did not show crack formation after CR.

**Figure 3 materials-12-01526-f003:**
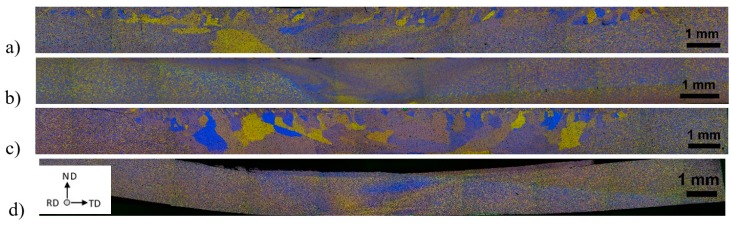
Polarized optical micrograph (POM) Montage of the FSW joint sections of the four experimental conditions of annealing (A), FSW, and CR: (**a**) FSW_1_-A-CR; (**b**) A-FSW_1_-CR; (**c**) FSW_2_-A-CR; and (**d**) A- FSW_2_-CR.

**Figure 4 materials-12-01526-f004:**
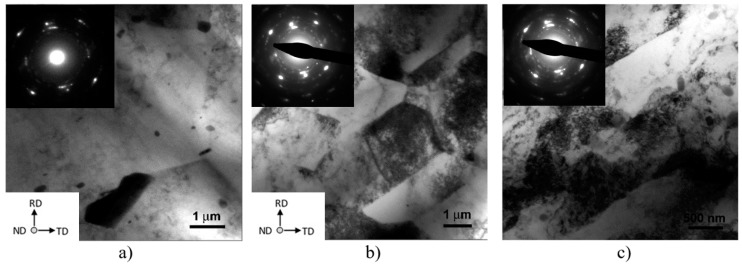
TEM representative micrographs of the nugget zone of (**a**) FSW_2_, (**b**) FSW_2_-CR (*ε* = 1.08), and (**c**) ECAP-1 (*ε* = 1.08) inspected in the X-plane. RD, TD, and ND were the same for all the three micrographs. In all three cases, Selected-Area Diffraction Patterns (SAEDPs) were recorded with equal aperture size of 800 nm and same Al-[002] crystallographic orientation. In (**c**), the ECAP pressing direction is horizontal, from left to right.

**Figure 5 materials-12-01526-f005:**
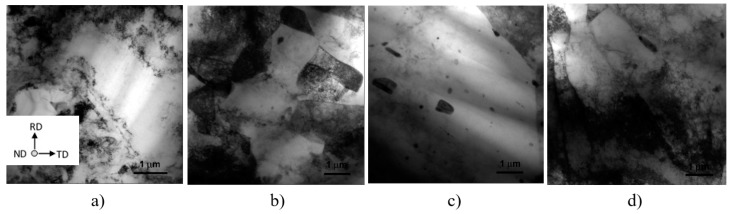
(**a**) FSW_1_-A-CR, (**b**) A-FSW_1_-CR, (**c**) FSW_2_-A-CR, and (**d**) A-FSW_2_-CR. All TEM micrographs refer to the FSW nugget zone. RD, TD, and ND were the same for all the four micrographs.

**Figure 6 materials-12-01526-f006:**
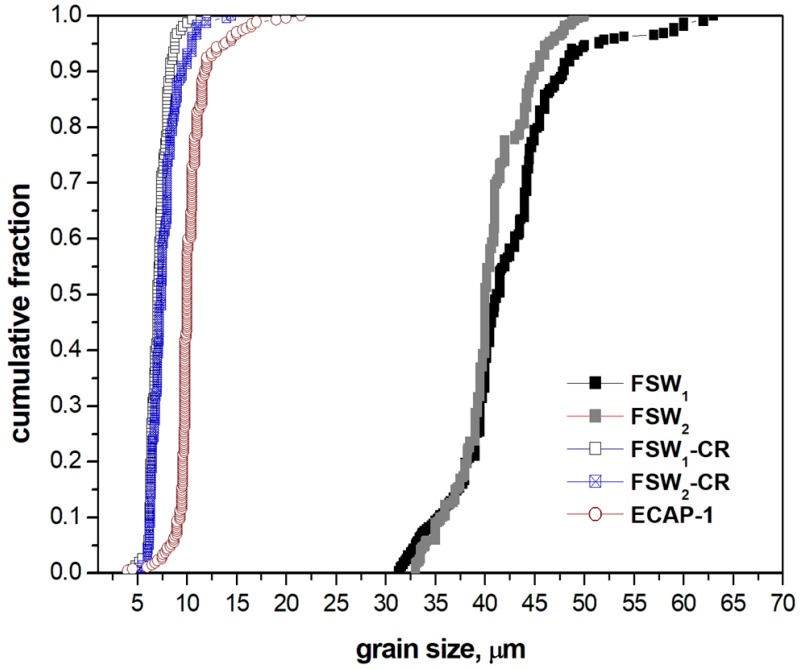
Grain size distribution of FSW_1_, FSW_2_, FSW_1_-CR, FSW_2_-CR, and ECAP-1.

**Figure 7 materials-12-01526-f007:**
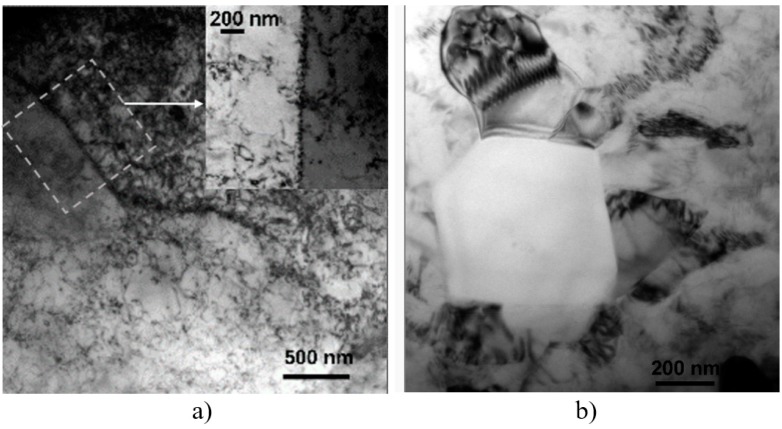
(**a**) Dislocation wall formation, and (**b**) subgrain formation, within the NZ of the A-FSW_2_-CR condition.

**Figure 8 materials-12-01526-f008:**
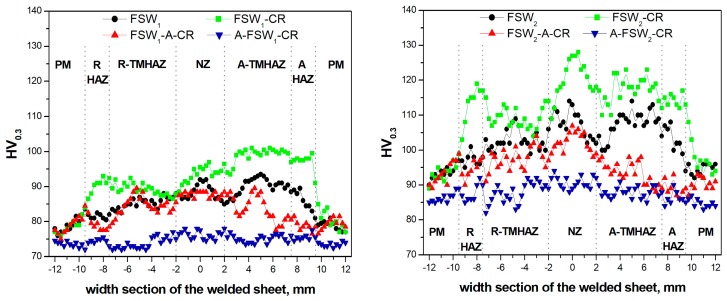
Microhardness profiles across the welded zone sections in the FSW, FSW-CR, and FSW-A-CR, A-FSW-CR, for *ω* = 600 rpm/*v* = 200 mm/min (FSW_1_, left-hand side), and *ω* = 1800 rpm/*v* = 75 mm/min (FSW_2_, right-hand side).

**Figure 9 materials-12-01526-f009:**
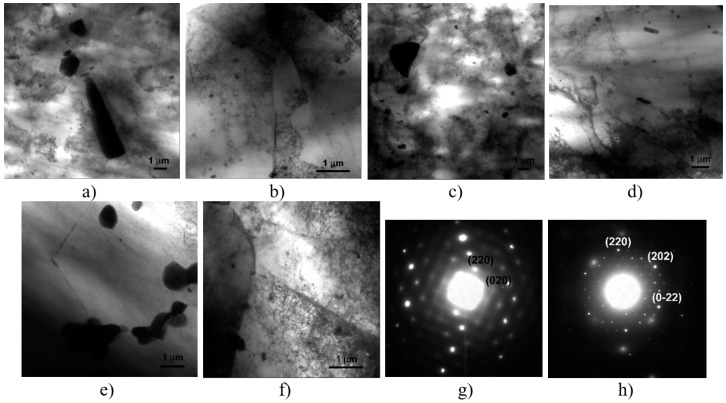
MnAl_6_ dispersoids, Mg_2_Al_3_ particles, and (Fe,Mn)_3_SiAl_12_ intermetallic particles evolution with the different experimental conditions: (**a**) FSW_1_-A-CR, (**b**) A-FSW_1_-CR, (**c**) FSW_2_-A-CR, (**d**) A-FSW_2_-CR, and (**e**), (**f**) ECAP-1. (**g**) SAEDP of the MnAl_6_ dispersoids, taken at the [001] matrix zone axis, and (**h**) SAEDP of the Mg_2_Al_3_ particles taken at the [111] matrix zone axis. MnAl_6_ was identified according to the JCPDS chart #6-665, as orthorhombic phase (S.G. C_MCM_/63) with a = 6.50 Å, b = 7.55 Å, c = 8.87 Å; Mg_2_Al_3_ was identified according to the JCPDS chart #3-877, as cubic phase (S.G. C_P3M_/227) with a = 28.28 Å.

**Table 1 materials-12-01526-t001:** Experimental conditions tested in the present study and the sequence of the three different treatments to which the AA5754 was subjected (from the first, I, the second, II, and to the third, III).

Experimental Condition Sequence	*ω_1_* = 600 rpm *v_1_* = 200 mm/min	*ω_2_* = 1800 rpn *v_2_* = 75 mm/min	Annealing (A) 415 °C/3 h	CR Deformation *ε* = 1.08
FSW_1_-CR	I	-	-	II
FSW_2_-CR	-	I	-	II
FSW_1_-A-CR	I	-	II	III
FSW_2_-A-CR	-	I	II	III
A-FSW_1_-CR	II	-	I	III
A-FSW_2_-CR	-	II	I	III

**Table 2 materials-12-01526-t002:** Mean grain, d_g_, and cell (subgrain), *d_c_*, sizes.

Experimental Condition	*d_g_* (μm)	*d_c_* (nm)
FSW_1_	40 ± 3	-
FSW_2_	42 ± 5	-
FSW_1_-CR	7 ± 1	610 ± 90
FSW_2_-CR	8 ± 1	690 ± 80
ECAP-1	10 ± 2	1000 ± 100
FSW_1_-A-CR	6 ± 1	800 ± 100
FSW_2_-A-CR	1.1 ± 0.4	1100 ± 200
A-FSW_1_-CR	2.8 ± 0.6	500 ± 100
A-FSW_2_-CR	3.6 ± 0.6	600 ± 100

**Table 3 materials-12-01526-t003:** Microhardness mean values in the nugget zone for the different experimental conditions: FSW_1_, FSW_2_, FSW_1_-CR, FSW_2_-CR, FSW_1_-A-CR, FSW_2_-A-CR, A-FSW_1_-CR, A- FSW_2_-CR, and ECAP-1.

Experimental Condition	H, *HV_0.3_*
FSW_1_	86 ± 4
FSW_2_	103 ± 5
FSW_1_-CR	93 ± 4
FSW_2_-CR	113 ± 6
FSW_1_-A-CR	83 ± 4
FSW_2_-A-CR	95 ± 5
A-FSW_1_-CR	74 ± 2
A-FSW_2_-CR	88 ± 3
ECAP-1 (*ε* = 1.08)	98 ± 4

**Table 4 materials-12-01526-t004:** Quantitative evaluation of the MnAl_6_ dispersoids, Mg_2_Al_3_ particles, and (Fe,Mn)_3_SiAl_12_ intermetallic particles.

Experimental Condition	V_V_(MnAl_6_), ‰	V_V_(Mg_2_Al_3_), ‰	d_p_(MnAl_6_), nm	d_p_(Mg_2_Al_3_), nm
FSW_1_-CR	50 ± 15	10 ± 4	80 ± 15	150 ± 10
FSW_2_-CR	50 ± 15	8 ± 3	85 ± 15	150 ± 15
FSW_1_-A-CR	80 ± 15	18 ± 6	85 ± 15	160 ± 20
FSW_2_-A-CR	80 ± 10	28 ± 8	80 ± 15	160 ± 20
A-FSW_1_-CR	75 ± 15	12 ± 6	90 ± 25	170 ± 30
A-FSW_2_-CR	70 ± 15	14 ± 6	90 ± 25	170 ± 30
ECAP-1 (ε = 1.08)	60 ± 10	12 ± 4	80 ± 10	160 ± 20

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
