# Peer review of "Post-FSW Cold-Rolling Simulation of ECAP Shear Deformation and Its Microstructure Role Combined to Annealing in a FSWed AA5754 Plate Joint"

_materials, 2019, doi:10.3390/ma12091526_

Round 1
Reviewer 1 Report
The manuscript consists of a number of interesting findings on structure and properties of friction stir welded and cold rolled AA5754 alloy and therefore can be recommended for publication. However there are some comments and questions which should be addressed.
1. The occurrence of shear deformation (and reasons of that) during rolling should be described/explained more clear. Was it due to different circular velocity or different diameters of the rollers, for example?
2. Tabulated grain and subgrain sizes would be more convenient for understanding.
3. For TEM images SAED patterns would be useful for supporting some statements on grain- or subgrain structure.
Author Response
see attached word document.

Reviewer 2 Report
In the article authors investigate the effect of combined friction stir welding process and cold rolling on the microstructure and hardness of AA5754 joint and compare the observations with ones obtained for the same alloy processed by a single pass of ECAP.
The presented research material is interesting and relevant from the point of view of technological application as well as scientific standpoint, however there are problems with the manuscript that authors should alleviate before the research could be published:
1) There is a significant discrepancy between title of the manuscript, the abstract and the contents of the manuscript main body. As the title claims the research is focused on the applied processing route and its ability to simulate grain refinement obtained in ECAP. However a significant part of the results and discussion, as well as almost half of conclusions sections are devoted to the influence of applied annealing on the size and volume of intermetallic precipitates. Furthermore, those results are also not mentioned in the abstract at all.
As the mentioned results take up a considerable portion of the manuscript, in my opinion the title and abstract should be amended to include appropriate information, and to better reflect contents of the paper.
2) In the work, authors used a custom cold rolling setup to simulate strain obtained in a single ECAP pass. Authors claim that used CR setup provides equivalent strain of ε = 1.08, however there is no information on how exactly this value was estimated. Information on the method of strain value estimation should be added.
3) There is very little information on the geometry of the CR setup, which makes it difficult to judge how this processing method actually compares to ECAP. In ECAP the main deformation mode is shearing in the cross section plane of two channels, the amount of shear depends on intersection angle and curvature angle. It’s not clear how the presented CR setup recreates this conditions. Authors should provide information on how the shearing is obtained in their setup as well as what is rollers diameter, what is the value of ΔR in fig1, what is the rolling speed, is there a roller speed asymmetry?
4) Description of directions (RD, TD, ND) should be introduced in the fig 1 and carried out throughout of the manuscript, especially in the figs 2 and 3.
5) In the manuscript the grain refinement effect of CR is compared to grain refinement effect of ECAP, however there is no information on what the grain size was before the ECAP or before the combined FSW-CR process, this information should be added.
6) In this work statistical microstructure parameters such as grain size and cell size are investigated using visible light microscope and TEM. Why the EBSD technique was not used? The reported mean grain values from about 1 to about 6 micrometres after combined FSW and CR seem suitable for this kind of analysis. How many TEM images was used to determine average grain size values, what was total analysed area per sample? To the best knowledge of the reviewer it is often not possible to distinguish between low angle and high angle grain boundaries based on TEM bright or dark field observations. How this was performed here?
7) It would be beneficial to readability of the manuscript if the microstructure parameters (grain size, cell size) were presented graphically or in a table instead listing them in the text.
8) Is the grain size distribution affected by different processing routes, can authors provide histograms of grain size distributions?
9) Overall the quality of English writing is ok, however the punctuation should be carefully reviewed (especially unnecessary commas), and a reading by a native speaker is advised. Some additional suggested language changes: a phrase ‘respect to’ is used incorrectly a number of times in the text, a correct one is ‘with respect to’ (see: https://www.ldoceonline.com/dictionary/with-respect-to-something), also in line 407: ‘FSW, compared to the way round.’ should be ‘compared to the other way (a)round’ (see: https://www.ldoceonline.com/dictionary/the-other-way-around-round), ‘grain structure’ should be used instead of ‘grained structure’.
Author Response
see attached word document.

Reviewer 3 Report
In this study, a plastic deformation process of cold rolling (CR) on a FSWed sheets was tested with the goal of obtaining a grain refinement . The paper was well structured and analyzed. I just wondering that how did you measured the hardness, by using the nanoindentation or other method? please give the experimental conditions. In addition. if the author can provide the grain size after refine, that would be better.
Author Response
see attached word document.
